# Chronic exercise effects on overall depression severity and distinct depressive symptoms in older adults: A protocol of a systematic and meta-analytic review

Melanie Mack[1,2]*, Andreea Badache[3], Arzu Erden[4], Christoforos D. Giannaki[5], Sandra Haider[6], Antonia Kaltsatou[7], Burcu Kömürcü Akik[8], Yaël Netz[9,10], Iuliia Pavlova[11], Pinelopi S. Stavrinou[5], Claudia Voelcker-Rehage[2], Michel Audiffren[12], on behalf of the PhysAgeNet[¶]

1 Centre for the Interdisciplinary Study of Gerontology and Vulnerability, University of Geneva, Geneva, Switzerland, 2 Department of Neuromotor Behavior and Exercise, Institute of Sport and Exercise Sciences, University of Münster, Münster, Germany, 3 Department of Disability Research, National Research School on Ageing and Health, Örebro University, Örebro, Sweden, 4 Department of Physiotherapy and Rehabilitation, Faculty of Health Science, Karadeniz Technical University, Trabzon, Turkey, 5 Department of Life Sciences, University of Nicosia, Nicosia, Cyprus, 6 Center for Public Health, Department of Social and Preventive Medicine, Medical University of Vienna, Vienna, Austria, 7 Functional Architecture of Mammals in their Environment (FAME) Laboratory, Department of Physical Education and Sport Science, University of Thessaly, Volos, Greece, 8 Department of Psychology, Faculty of Languages and History-Geography, Ankara University, Ankara, Turkey, 9 The Levinsky-Wingate Academic Center, Tel Aviv, Israel, 10 Department of Health Promotion and Rehabilitation, Lithuanian Sports University, Kaunas, Lithuania, 11 Department of Theory and Methods of Physical Culture, Lviv State University of Physical Culture, Lviv, Ukraine, 12 Cognition and Learning Research Center, Maison des Sciences de l'Homme et de la Société, CNRS, University of Poitiers, Poitiers, France

¶ The complete membership list of the consortium is available here: https://www.cost.eu/actions/CA20104/#tabs+Name:Working%20Groups%20and%20Membership
* melanie.mack@unige.ch

**Data Availability Statement:** No datasets were generated or analyzed during the current study. All relevant data from this study will be made available

## Abstract

### Introduction

There is high evidence that chronic exercise benefits overall depression severity in older adults. However, late-life depression is characterized by considerable heterogeneity in clinical manifestation emphasizing the need for more individualized exercise intervention programs. Therefore, the objective of the proposed review is to investigate the effects of chronic exercise on overall depression severity and on different symptoms of depression in randomized controlled trials (RCTs) including older adults with a mean age of at least 60 years, and by considering the moderating effects of intervention characteristics and individual characteristics.

### Methods

This protocol is guided by the Preferred Reporting Items for Systematic Reviews and Meta-Analysis Protocols (PRISMA-P). We will use the Population-Intervention-Comparator-Outcomes-Study design (PICOS) criteria for study inclusion and will search the following

upon study completion on Open Science Framework, OSF (DOI: 10.17605/OSF.IO/ZNWE8).

**Funding:** This publication is based upon work from EU COST Action CA20104 - Network on evidence-based physical activity in old age (PhysAgeNet), supported by COST (European Cooperation in Science and Technology). https://www.cost.eu/, https://physagenet.eu/. The funders did not and will not have a role in study design, data collection and analysis, decision to publish, or preparation of the manuscript. There was no additional external funding received for this study.

**Competing interests:** All the authors of this manuscript are members of the COST network PhysAgeNet (CA 20104). The authors have declared that no competing interest exists. This does not alter our adherence to PLOS ONE policies on sharing data and materials.

database sources for relevant RCTs: Web of Science, Academic Search Complete, CINAHL, APA Psycinfo, SPORTDiscuss, Cochrane. Two independent reviewers will conduct the study selection, data extraction, and quality assessment. Disagreement will be solved by a third reviewer. Primary outcome will be changes in overall depression severity and secondary outcomes will encompass changes in symptoms of depression as defined by the DSM-5, such as sleep quality, fatigue, anxiety, mood, apathy, changes in weight, information processing speed, and executive functions, from baseline until the end of the intervention and to any available intermediary measurement or follow up. Meta-analysis will be undertaken to synthesize the effects of chronic exercise on primary and secondary outcomes. Subgroup analysis will investigate the moderating effects of intervention characteristics (frequency, intensity, duration, type of exercise, cognitive demand, social interactions, exercise supervision, behavioral change techniques, compliance, study design, dropout-rate, type of control group) and individual characteristics (age, sex, education, functional capacity, global cognition, population) on primary and secondary outcomes. Additionally, we plan to assess quality of evidence and publication bias, and to carry out sensitivity analysis.

## Conclusion

The results of the proposed review are anticipated to have a substantial impact on research and clinical practice. On the one hand, the review's conclusions could form the foundation for developing evidence-based recommendations for individualized exercise programs that alleviate depression in older adults. On the other hand, by revealing research gaps, the review results could encourage the formulation of research questions for further RCTs.

## Protocol registration number

This protocol has been published in the Prospero repository (PROSPERO 2022 CRD42022361418, available from: https://www.crd.york.ac.uk/prospero/display_record.php?ID=CRD42022361418)

## Introduction

Depression among older adults, also referred to as geriatric depression or late-life depression, presents a significant public health concern. The average global prevalence of geriatric depression, including estimates assessed between 1994 and 2020, is 31.74% [1], and rates of depressive disorders increase substantially in the highest age group 85 years and older, reaching up to 50% [2]. These high numbers may be attributed to various factors associated with aging, such as a higher proportion of women, more physical disability, higher cognitive impairment, and lower socioeconomic status [3, 4]. Given its high prevalence and potential negative consequences, such as increased physical illness, impaired functioning, suicide risk, and reduced quality of life, depression in older adults deserves special attention [5, 6].

Beneath the symptoms of depression described in the American Psychiatric Association's Diagnostic and Statistical Manual (DSM-5 [7]) and in the International Classification of Diseases (ICD-10 [8]), late-life depression is especially associated with cognitive and somatic changes [6, 9]. Cognitive changes may manifest as difficulties with concentration, speed of

information processing, and executive functions [10, 11], while somatic changes may include increased abdominal fat, type 2 diabetes, and hypertension [12]. In addition to medical and neurological comorbidity [13, 14], late-life depression is frequently associated with disability and psychosocial difficulties. These health domains often interact with one another, contributing to the development of geriatric depression [13, 15]. For instance, cerebrovascular disease may cause depression and cognitive impairment, which can lead to disability and social isolation, further worsening depressive symptoms [16, 17]. Once depression sets in, it may increase cognitive impairment, disability, psychosocial problems, and morbidity. Furthermore, older individuals with recurrent depression since younger age and those with late-life onset seems to have distinct clinical features, risk factors, and treatment responses [9, 18]. Compared with older patients with early-onset depression, patients with late-onset depression appear to have a higher prevalence of medical burden and neurological brain disorders [19], more impairment in neuropsychological tests [20] and dysfunctions of several brain regions [9, 20].

Given the complexity and heterogeneity of late-life depression, it is not surprising that many patients are not adequately treated [21]. The first line of treatment of late-life depression is commonly pharmacotherapy. However, antidepressants pose a considerable risk for side effects, especially in the presence of comorbidities and the administration of adjunctive medications, including risk of falls, osteoporosis, as well as cardiac and sexual dysfunction [21]. Thus, the majority of patients receive no treatment, an insufficient duration of treatment, or a lower than recommended dose [22, 23]. In addition, antidepressant efficacy in older adults is modest and declines with age, which may contribute to the higher rates of depression relapse in this population [21]. In recent years, regular physical exercise has gained attention in clinical practice and research as a promising noninvasive treatment option for late-life depression. By the majority of populations, exercise can be performed autonomously without additional costly equipment or external assistance, has no serious side effects, and is accessible to everyone.

Regular physical exercise, often termed as chronic or systematic exercise, is defined as planned, structured, and repetitive bodily movements done to improve or maintain one or more components of physical fitness [24]. Interventions including exercise aiming to reduce depression can be referred to as non-pharmacological interventions or behavioral medicine/ therapy interventions and may fall within the scope of evidence-based behavioral medicine or behavioral therapy [25]. Evidence-based behavioral medicine is a systematic approach to use the best available scientific evidence from clinical research to help make decisions about the care of individual patients [26]. This approach has become particularly important in the development of clinical standards and guidelines to improve the quality of care. Systematic reviews of the available published evidence are required to identify interventions that result in improvements of behavior and health. The validity and explanatory power of reviews depends on the quality of the published research, but also on the compilation and descriptions of the aspects that are important for care [27–29]. The extension of the CONSORT statement for RCTs of non-pharmacological interventions specified a list of relevant aspects to be reported, such as the precise description of both the experimental treatment and the comparator, the implementation of the intervention, and how adherence was promoted and assessed [27]. In the following, we will discuss which aspects might be especially important regarding exercise interventions for late-life depression.

## Intervention characteristics

Exercise interventions benefit depressive symptoms in older adults by promoting a variety of psychological and physiological mechanisms relevant to late-life depression. The former

includes psychosocial and cognitive factors, such as augmenting self-esteem [30], the latter includes anti-inflammatory effects or neuroplasticity mechanisms, beyond others [31, 32]. However, the heterogeneity of the effects is high, which might be caused by variations between exercise interventions. The induced mechanisms and their strength depend strongly on the specific training regime. One important aspect that impacts the effects of exercise interventions are the FITT principles of training, which stands for 'Frequency', 'Intensity', 'Time', and 'Type of exercise' [24]. 'Frequency' refers to the number of exercise sessions per week, while 'Intensity' refers to the level of exertion during exercise. 'Time' refers to the duration of each exercise session, and 'Type' refers to the specific mode of exercise, such as aerobic, resistance, or mind-body exercises. By manipulating the four components of the FITT principles, exercise programs can be designed to elicit specific adaptations in the body. For example, aerobic exercises or resistance exercises have a high metabolic demand that benefit depression by increasing brain volume (gray and white matter of primarily prefrontal and temporal regions) or peripheral biomarkers, such as BDNF [33]. Mind-body exercises, such as Yoga, Tai Chi, balance and other coordinative exercises, place high demands on the neuromuscular and neurocognitive system that might benefit depression by initiating functional and structural changes [34]. The effect size of exercise interventions is further modulated by frequency, intensity, and duration of exercise sessions [35]. Exercises with a higher intensity might induce higher acute neuroendocrine responses and long-term adaptations in the brain's neural architecture than exercises with lower intensity [36]. Meaningful responses may be only visible when the exercises are conducted with a certain frequency and duration [37].

Beneath the FITT principle, further important components of exercise interventions to improve late-life depression are psychosocial aspects, such as the degree of social interaction among participants or the use of behavioral change techniques. Social interaction differs between group-based and individual-based (including home-based) interventions but also within group-based interventions. Group-based training sessions, for example, may include exercises with partner(s) in cooperation and/or opposition or only verbal interactions [38]. The extent of social interaction can influence social connectedness and thus social identity and loneliness. Loneliness, defined as feelings of isolation, disconnectedness, and lack of belonging, is a strong and well-established risk factor for depression [39]. The social identity approach assumes that belonging to a group can protect against depression because of the associated access to resources (e.g., social support from other members), even in the absence of access to support from or contact with other group members [40, 41]. Behavioral change techniques, such as goal-setting, self-monitoring, and problem-solving, are strategies that are used to promote behavior change. In the context of exercise interventions, behavioral change techniques may be used to help individuals set and achieve realistic exercise goals, track their progress, and overcome barriers to exercise [42]. Parts of those techniques are also included in cognitive behavioral therapy programs that are known to have beneficial effects for older adults suffering from depression [43]. Although in exercise interventions, these techniques are applied to the context of exercise behavior, it is plausible that they can also be translated to other areas of life, such as dealing with depression. One might for example assume that participants who learn to make connections between their thoughts, feelings, and actions in relation to activity behavior can also transfer this strategy to depression and develop alternative ways of coping with the depressive symptoms.

RCTs are considered the gold standard for effectiveness research because they allow a cause-effect relationship to be established between the intervention and the outcome. However, for exercise interventions with psychosocial or behavioral components, it is difficult to identify appropriate control groups because the active/inactive components of the interventions are not as obvious as they are for medication interventions. This might lead to

methodological issues in the choice of the control group that might influence the results [44]. For example, an RCT including an active control group with social interactions to examine the effect of an individual-based exercise intervention (without social interaction) is not appropriate. Therefore, the choice of control group in RCTs on exercise interventions for late-life depression should be considered when interpreting results.

## Individual characteristics

In attempting to generate reliable and clinically relevant evidence on the effects of exercise in late-life depression, the question of "what works for whom," also called precision medicine or personalized medicine is important to be considered. In this vein, the choice of the outcome but also the population to be treated might also be of importance. Late-life depression can be assessed with different validated depression scales, such as the Beck Depression Inventory-II (BDI-II [45]) or the Geriatric Depression Scale (GDS [46]). These standardized clinical instruments typically quantitatively measure the severity of depressive symptoms in a single score based on a multidimensional assessment of a range of depressive symptoms, including mood, appetite, sleep, energy, and concentration, as well as feelings of worthlessness, hopelessness, and guilt. By providing one overall depression score, depression rating scales can be helpful in clinical decision-making, tracking treatment progress, and determining the effectiveness of various interventions. Nevertheless, one single score provides only limited information on the severity of the specific symptoms that differ enormously in late-late depression, and which are linked to specific pathophysiological mechanisms as described above. Considering that exercise affects the brain and body by specific pathways, it is plausible that the effects of exercise on depression may vary depending on whether those pathways are impaired or vulnerable in the individual patient.

Older adults suffering from depression further differ concerning various individual characteristics, such as age, sex, physical fitness, cognitive functions, years of education and comorbidities. It is important to note that these individual characteristics might influence which exercise intervention is appropriate and leads to the most beneficial effects [47]. For instance, the physiological response to equivalent 'dosages' of exercise differs between males and females [48]. Therefore, gender might influence the intervention outcomes. Different comorbidities might also lead to differential effects of exercise. First, exercise is not without risk. Potential adverse events include cardiac events or hypoglycemia, and to ensure the safety of exercise certain precautions, such as reducing the intensity, or the duration of an exercise session, are necessary for certain populations. These precautions might imply that the impact of exercise is insufficient for reducing depression. However, it is also possible that exercise has a positive effect not only on depression but also on comorbid conditions. If these comorbidities are linked to depression (for instance, frailty leading to depression as a result of the loss of the ability to carry out daily activities), it could potentially result in even greater improvement in depression. Although these connections have not been investigated yet, they hold significance in the context of geriatric depression.

## State of research

Several systematic reviews and meta-analyses investigated the effect of a large variety of chronic exercise interventions on depression in various populations of older adults (for an overview see umbrella reviews [49, 50]). Most of those overview articles on the effects of exercise on late-life depression target only overall depression severity as an outcome variable of interest, assessed with different validated depression scales [51–58]. Although there are quite many RCTs examining the effects of chronic exercise on specific symptoms of depression (to

name only a few [59–61]), to our knowledge, there is no published review or meta-analysis that attempts to uncover the effects of chronic exercise on the broadest possible range of symptoms as described in the DSM. However, given the substantial heterogeneity in clinical presentation of late-life depression, it underscores the need of customizing the exercise interventions to the unique symptom expressions of each patient.

Overall, already conducted reviews and meta-analysis suggest a moderate effect of chronic exercise on the severity of late-life depression, with the exercise group showing a higher decrease of depression severity compared to a control group with no exercise intervention [49]. A proportion of those overview articles also examined the effects of different moderator variables on this exercise-depression relationship. The studies addressed exercise intervention characteristics, such as intensity of exercise [47], type of exercise [47, 54], type of control group [53]; but also, individual characteristics, such as age [51, 56, 62], comorbidities [47, 62, 63], severity of depression [62, 64], or cognitive impairment [62]. The results show, on the one hand, a high degree of heterogeneity for the exercise characteristics and, on the other hand, partly different results for the individual characteristics. One example is the moderating effect of age. While one study found an increasingly beneficial effect of exercise with increasing age [51], another study found opposite results with decreasing positive effects with increasing age [62], and a third study found no moderating effect of age [56]. One reason for the varying and heterogeneous results could be that the studies included in the reviews differ in various aspects of individual and exercise characteristics. For example, the overview articles mentioned above differ in the type of exercises included: only mind-body exercises [51] versus only aerobic and resistance exercises [62] versus all three types of exercises [56]. It might be suggested that exercises, which vary in their characteristics, have different effects on depression due to the distinct neurobiological and psychological mechanisms they involve [54, 65, 66]. Thus, they may also have differential effects in different populations. For example, one might assume that moderate-intensity physical exercise with high cognitive demand is particularly beneficial for the elderly, as the likelihood of cognitive decline increases with age. Unfortunately, there is no review that comprehensively examines the influence of potentially relevant moderator variables on the relationship between chronic exercise interventions and depression in older adults.

In addition, there are numerous sources of bias that might threaten the validity of meta-analyses, such as low methodological quality or publication bias. In a recent umbrella review on the effects of chronic exercise on depression in older adults, only one of the twelve included reviews was considered to be of high methodological quality as defined by the rating of the 'A MeaSurement Tool to Assess systematic Review– 2' (AMSTAR-2 [67]) [49]. In terms of publication bias, there is evidence that previous meta-analyses on the effects of chronic exercise on depression have underestimated the benefits of exercise due to publication bias [57, 68]. This is different than for other treatments, such as psychotherapy whose effect sizes have consequently been overestimated [69], and would speak for an even higher beneficial effect of exercise on late-life depression as suggested by now. Since publication bias is seldom calculated, further evidence is needed to strengthen the evidence.

## Rationale and objective

To sum up, available reviews provide evidence for the beneficial effect of chronic exercise on depression severity in older adults, but several issues remain unaddressed: (1) while most of the literature uses overall depression severity as an outcome, it is unclear whether chronic exercise differentially affects the various symptoms of depression; (2) the numerous reviews that have already examined the moderating effects of intervention characteristics and

individual characteristics have unfortunately found heterogeneous and partly differential results, probably because of their differential inclusion criteria with respect to those characteristics (e.g., only including mind-body interventions vs. only aerobic interventions or only including participants with neurodegenerative disease vs. only participants with major depressive disorder). Consequently, the moderating role of those characteristics and their interaction is still not fully understood; (3) low methodological quality of the included RCTs and publication bias seem to influence the validity and interpretation of the results but are only rarely considered. To derive evidence-based treatment decisions, reviews considering those aspects are necessary.

We hypothesize that chronic exercise interventions differ in their efficacy in reducing overall depression severity and the different symptoms of depression in older adults across participants with different individual's characteristics (i.e., age, sex, population, level of education at baseline, functional capacity at baseline, global cognition at baseline, depression severity at baseline) and across interventions with different exercise characteristics (i.e., frequency of exercise sessions, intensity of exercise, duration of exercise session, duration of intervention, type of exercise, cognitive demand of exercise, social interaction, use of behavioral change techniques, type of control group). Therefore, this protocol outlines a review aiming to investigate how the above-mentioned intervention characteristics and individual characteristics moderate the effects of chronic exercise on overall depression severity and the different symptoms of depression in older adults. The different symptoms of depression will be thereby defined using the criteria delineated by the DSM-5 [7] to make a diagnosis of depression: depressed mood, loss of interest/pleasure, weight loss or gain, insomnia or hypersomnia, psychomotor agitation or retardation, fatigue, feeling worthless or excessive/inappropriate guilt, decreased concentration, thoughts of death/suicide. Each of these symptoms will be used as secondary outcome in the present meta-analysis. In addition, we aim to assess quality of evidence and to consider publication bias [70].

The review will differ from previous reviews in several important ways: (1) by including a large sample of studies with a wide range of individual and exercise characteristics, the chronic effects of exercise on late-life depression will be examined as a function of a large variety of moderators related to these characteristics; (2) the effects of chronic exercise will be examined not only on the level of overall depression severity but on the level of the different symptoms of depression; (3) the results will be corrected for publication bias with different complementary techniques.

## Method

This protocol has been published in the Prospero repository (PROSPERO 2022 CRD42022361418, available from: https://www.crd.york.ac.uk/prospero/display_record.php?ID=CRD42022361418) and follows the Preferred Reporting Items for Systematic Reviews and Meta-Analyses Protocol (PRISMA-P) statement (see S1 Checklist) [71].

### Eligibility criteria

We will use the Population-Intervention-Comparator-Outcome-Study design (PICOS) criteria to determine primary study inclusion.

### Population

To be considered in this review, studies must fulfill the following criteria: (1) age: the mean age of each arm of the RCT is 60 years or above and the minimal age of each arm of the RCT is 50 years (i.e., the mean age minus standard deviation of age must be 50 years or above); (2) level

of depression: the mean level of depression of each arm of the RCT is mild / moderate or higher as attested by any validated scale assessing depression (i.e., the mean score of depression + the standard deviation must be ≥ cutoff value). For each depression scale we will therefore define the cutoff value based on the literature. The list of all validated scales and corresponding cutoffs will be made available on Open Science Framework (OSF). Our review further aims to encompass all populations of older adults independent of disease (e.g., dementia, cardiovascular disease, etc.), physical fitness and cognitive functioning or setting (e.g., clinical setting, community setting, etc.).

## Intervention

To be eligible for this review, studies must include at least one intervention arm comprising a chronic exercise component defined as any kind of "planned, structured, and repetitive bodily movement done to improve or maintain one or more components of physical fitness" [24], such as aerobic exercise, resistance exercise, motor-coordinative exercise, mind-body exercise, exergames, hybrid exercise, multicomponent exercise (also including cognitive training) and concurrent training. The intervention must last at least one week and include more than one exercise session. Intervention arms comprising a chronic exercise component additionally to other intervention components, such as nutrition, cognitive behavioral therapy, or pharmacotherapy, are not eligible. However, this only applies if the cognitive behavioral therapy or pharmacotherapy component is specifically manipulated and tested in the study. If they are part of the usual treatment, the study will be included. Furthermore, we will only include studies in which frequency of exercise sessions, duration of exercise sessions, duration of the intervention and type of exercise are reported.

## Comparator groups

Studies included in this review must include either a passive or an active control group. Eligible passive control groups include: (1) usual care / treatment as usual (TAU): the diseased participants continue to receive their usual treatment; (2) habitual activities: the healthy participants do not change their life habits; (3) waiting list: the participants do not change their life habits during the duration of the intervention but then follow the tested treatment. Eligible active control groups include: (1) health education: the participants follow a health education program; (2) non-physical leisure activities: the participants practice non-physical leisure activities, such as listening to music, reading newspapers, magazines or books, watching documentaries or movies, handicraft, playing games, etc.; (3) stretching / very light intensity exercise / relaxation: the participants practice exercises with a very light intensity, such as stretching, moving joints, relaxation; (4) social interactions: the participants communicate regularly with other participants.

## Outcomes

Studies eligible for this review must include a measure of depression scores at baseline and at the end of the intervention, as measured by a validated scale (c.f., Population).

## Study design

Eligible studies only include RCTs with either a parallel-group design (each participant is randomly assigned to one of multiple study arms) or a cluster design (pre-existing groups of participants, such as villages, nursing homes, are randomly assigned to one of multiple study arms).

### Article type

For inclusion in this review, a study must be a peer-reviewed scientific report or a PhD thesis of an RCT written in English language. Abstracts from congresses and study protocols are not considered.

### Information sources and search strategy

The following electronic databases will be searched from inception to shortly before publishing the meta-analysis: Web of Science (all bases); Academic Search Complete, MEDLINE with full text, CINAHL with full text, APA Psycinfo, SPORTDiscuss with full text through EBSCO host; Cochrane. The complete search strategy can be found in the S1 File. In addition, reference lists of the included studies will be manually searched to identify additional relevant studies (e.g., from previous meta-analyses on the same topic). The search will be re-run prior to the final analysis for the months not covered by the initial search.

### Study selection

Study selection will be performed with the reference management system Rayyan [72]. Fig 1 illustrates the planned study selection process. The results of literature searches will be imported into Rayyan using RIS format, and duplicates will be removed. The selection of studies will be conducted in two steps. Pairs of two reviewers will be involved in the study selection process. In the first stage, the two reviewers will independently screen the title and abstract of

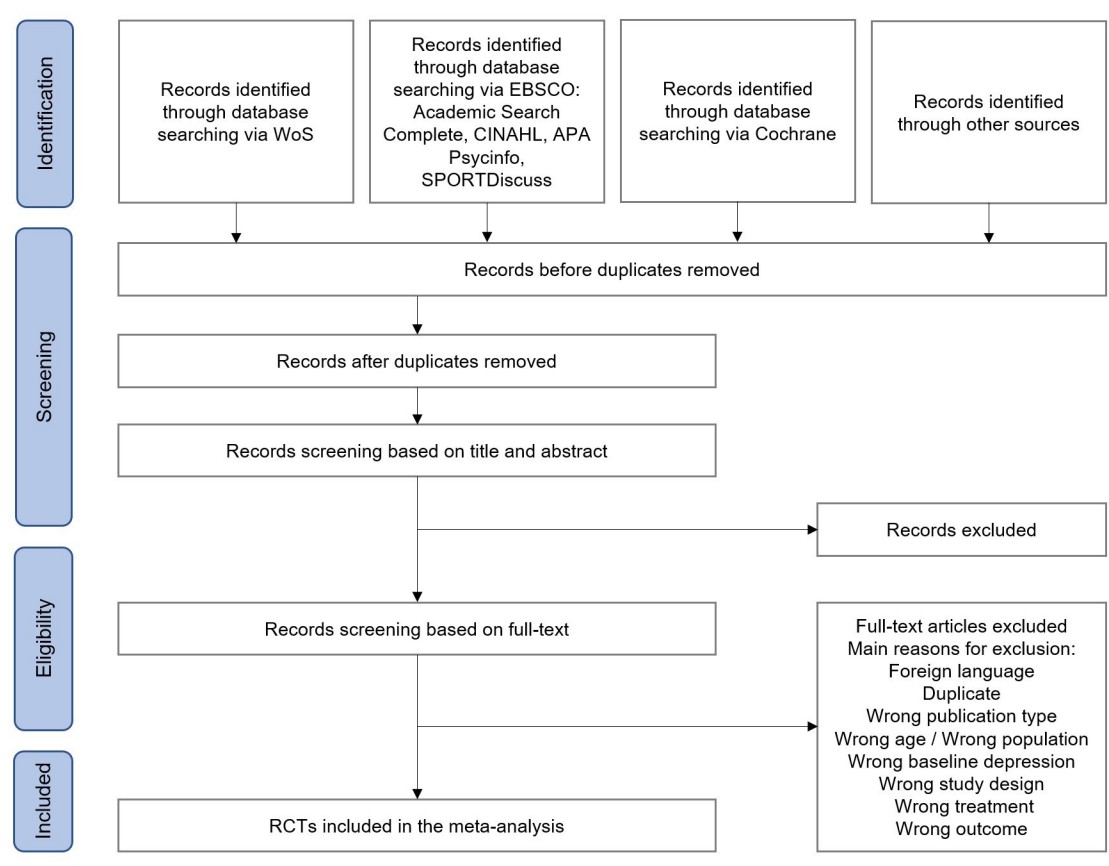

**Fig 1. Planned study selection process.**

their assigned studies. If a disagreement occurs between the two reviewers, a third reviewer will independently examine the eligibility criteria. Selected studies at stage 1 will then be assigned to other pairs of reviewers for the screening of the full-text articles. It follows the same procedure as the screening of the title and abstract. To ensure compliance with the work-flow and eligibility criteria, guidelines on the study selection process will be established for reviewers to follow and training sessions will be conducted. The studies selected in the different steps will be saved on Rayyan in respective directories. The guidelines for each stage of the selection process will be available on OSF.

## Data extraction and management

Study data will be collected and managed using REDCap electronic data capture tools hosted at Yale University [73, 74]. Pairs of two reviewers will be involved in the data extraction process. The two reviewers will independently extract data from included studies. If a disagreement occurs between the two reviewers, a third reviewer will independently examine the extracted data and correct the wrong data. The following pieces of information will be extracted by each pair of reviewers and entered in the REDCap database: first author of the report, year of publication, characteristics of the RCTs (i.e., number of arms, type of control group, etc.), primary outcomes, secondary outcomes, moderator variables and risk of bias. Outcomes and moderators are detailed hereafter. If data is missing, incomplete or questionable, investigators will be contacted to obtain or confirm data. To ensure compliance with the workflow, guidelines on the study selection process will be established for reviewers to follow; and training sessions will be conducted. The guidelines and the questionnaire used for the data extraction process will be available on OSF.

## Outcomes

All outcomes will be measured as the mean change of the values in the exercise group from baseline to the first post-intervention measurement time point, in comparison to the mean change of the control group. Therefore, we will extract the means and standard deviations from the exercise group and the control group at baseline and post-intervention measurement.

## Primary outcome

The primary outcome will be depression severity assessed with a validated depression scale. If an author reported the results of two primary outcome measures meeting our criteria (i.e., mean change in depressive symptoms according to two different measures), we will use both.

## Secondary outcomes

Secondary outcomes will include symptoms of depression delineated by the DSM-5 to make a diagnosis of depression. For the selection of the outcomes, we have adhered as closely as possible to each of the specific criteria, which are as follows: (1) depressed mood most of the day, nearly every day, as indicated by either subjective report (e.g., feelings of sadness, emptiness, hopelessness) or observation made by others (e.g., appears tearful); (2) markedly diminished interest or pleasure in all, or almost all, activities most of the day, nearly every day; (3) significant weight loss when not dieting or weight gain (e.g., a change of more than 5% of body weight in a month) or decrease or increase in appetite nearly every day; (4) psychomotor agitation or retardation nearly every day (observable by others, not merely subjective feelings of restlessness or being slowed down); (5) insomnia or hypersomnia nearly every day; (6) fatigue or loss of energy nearly every day; (7) feelings of worthlessness or excessive or inappropriate

guilt (which may be delusional) nearly every day (not merely self-reproach or guilt about being sick); (8) diminished ability to think or concentrate, or indecisiveness, nearly every day; (9) recurrent thoughts of death (not just fear of dying), recurrent suicidal ideation without a specific plan, or a suicide attempt or a specific plan for committing suicide. We have already completed a preliminary overview of RCTs on this topic to determine how many studies report each outcome of interest. If several instruments were available to assess a given symptom, we defined an order determined by the validity and frequency of use of the measurement instrument. An exception is the secondary outcome executive functions (see below). Here we will collect all reported measures. If not enough studies (less than three out of a total of approximately 100 studies) report a valid measure for a particular symptom, we will not assess that symptom as secondary outcome.

*Sleep quality* will be assessed with the following scales to measure aspects of depression related to the symptom "insomnia or hypersomnia nearly every day": (1) Pittsburg Sleep Quality index (PSQI) [75]; (2) Insomnia Severity Index (ISI) [76].

*Fatigue* will be assessed with the following scales to measure the symptom "fatigue or loss of energy nearly every day": (1) Fatigue Severity Scale (FSS) [77]; (2) Parkinson's Fatigue Scale (PFS-16) [78]; (3) Multidimensional Fatigue Inventory (MFI) [79]; and (4) Functional Assessment of Chronic Illness Therapy–Fatigue (FACIT-F) [80].

*Anxiety* will be assessed with the following scales to measure aspects of depression related to the symptom "depressed mood most of the day, nearly every day, as indicated by either subjective report (e.g., feels sad, empty, hopeless) or observation made by others (e.g., appears tearful)": (1) Beck Anxiety Inventory (BAI) [81]; (2) Generalized Anxiety Disorder—7 items (GAD-7) [82]; (3) Hamilton Anxiety Rating Scale (HAM-A) [83]; (4) Hospital Anxiety and Depression Scale, subscale Anxiety (HAS) [84]; (5) State-Trait Anxiety Inventory, subscale Trait Anxiety (STAI) [85]; and (6) Symptom-Checklist, subscale Anxiety (SCL-90-R) [86].

*Mood* will be assessed with the following scales to measure aspects of depression related to the symptom "depressed mood most of the day, nearly every day, as indicated by either subjective report (e.g., feelings of sadness, emptiness, hopelessness) or observation made by others (e.g., appears tearful)": (1) Alzheimer's Mood Scale, subscale Negative Mood [87]; (2) Dementia Mood Scale, subscale Mood (DMAS) [88]; (3) Positive and Negative Affect Schedule, subscale Negative Affect (PANAS) [89]; (4) Profile of Mood States (POMS) [90]; and (5) State-Trait Anxiety Inventory, subscale State Anxiety (STAI) [85].

*Apathy* will be assessed with the following scales to measure aspects of depression related to the symptom "markedly diminished interest or pleasure in all, or almost all, activities most of the day, nearly every day (as indicated by either subjective account or observation)": (1) Apathy Evaluation Scale (AES) [91]; (2) Neuropsychiatric Inventory Questionnaire, Apathy subscale (NIQ) [92]; and (3) Starkstein Apathy Scale (SAS) [93].

*Changes of weight* will be assessed with the Body Mass Index (BMI) to measure aspects of depression related to the symptom "significant weight loss when not dieting or weight gain (e.g., a change of more than 5% of body weight in a month) or decrease or increase in appetite nearly every day". We are aware that regular exercise might influence BMI changes and that this variable carries the potential for misinterpretation, as weight changes could mistakenly imply loss of appetite or unintended weight loss. Nonetheless, given the robust U-shaped association between overall depression severity and obesity status (i.e., underweight, normal weight, overweight / obese) [94], we consider this symptom to be of significant importance. It might be plausible that exercise leads to a normalization of BMI, with individuals with excessively high BMIs experiencing weight loss and those with excessively low BMIs gaining weight, rather than resulting in an overall reduction in BMI.

**Table 1. Overview of most used tests assessing executive functions in older adults.**

| Test or battery | Executive function | Outcome |
|---|---|---|
| Stroop task | Inhibitory control | Interference score, number of errors in incongruent trials |
| Random number generation task | Inhibitory control | Adjacency score |
| Hayling task | Inhibitory control | Inhibition score |
| Spatial Running Span task | Working memory | Number of correct responses |
| Verbal Running Span task | Working memory | Number of correct responses |
| 2-back task | Working memory | Number of correct responses |
| Dimension-switching task | Cognitive flexibility | Local switch cost |
| Plus-Minus task | Cognitive flexibility | Shifting cost |
| Digit-Letter task | Cognitive flexibility | Shifting cost |
| Counting backward | Working memory | Number of correct responses |
| Digit Span Backward | Working memory | Number of correct responses |
| Stop Signal task | Inhibitory control | Stop Signal Reaction Time |
| Attention Network Test [96] | Inhibitory control | Error rate |
| Wisconsin Card-Sorting Test [97] | Cognitive flexibility | Number of shifts correctly performed, perseverance errors |
| Frontal Assessment Battery [98] | Executive functions | Global score |
| Go / No Go task | Inhibitory control | Number of failed inhibitions |
| Trail making test [95] | Cognitive flexibility | TMTB score–TMTA score, TMTB score, TMTB / TMTA |

*Information processing speed* will be assessed with the following cognitive tests to measure aspects of depression related to the symptom "psychomotor agitation or retardation nearly every day (observable by others, not merely subjective feelings of restlessness or being slowed down)": (1) Trail Making Test, part A (TMT) [95]; and (2) reaction time (RT) tasks (simple RT task and two-choice RT task with compatible S-R mapping).

*Executive functions* will be assessed with the different cognitive tests to measure aspects of depression related to the symptom "diminished ability to think or concentrate, or indecisiveness, nearly every day" (see Table 1).

We will further consider changes in depression score and symptoms of depression from baseline to any available intermediary measurement or follow-up as secondary outcomes.

## Moderators

We will extract information about moderator variables that are either commonly used in related previously published meta-analyses and / or are hypothesized to modulate the effect of chronic exercise on depression.

## Intervention characteristics

Concerning exercise characteristics, we will extract information on the key components of the FITT principles:

- *Frequency of exercise sessions* assessed as mean number of sessions per week.

- *Intensity of exercise* categorized as follows: (1) light; (2) moderate; (3) vigorous / high [24]. The intensity of exercise will be determined by a dual approach utilizing both objective and subjective ratings. Objective ratings will be guided by cut-off points derived from literature for heart rate reserve (%HRR), oxygen uptake reserve (%VO$_2$R), maximal oxygen consumption (VO$_2$max), maximal heart rate (%HRmax), repetition maximum (%RM). Subjective ratings will be guided by predefined cut-off points for METs (metabolic equivalents of tasks),

ratings of perceived exertion (RPE), as well as specific exercise types classified by a Delphi survey by the COST action PhysAgeNet.

- *Duration of exercise sessions* assessed as mean duration of sessions in minutes.

- *Duration of the intervention* assessed as mean duration in weeks.

- *Type of exercise* categorized as follows: (1) aerobic exercise (e.g., Nordic Walking, swimming, running); (2) strength exercise / resistance exercise (e.g., weight-lifting, Pilates); (3) dance (Chinese scare dance, Turo, dance-based therapy, private ballroom dance, folk dance); (4) balance exercise / coordination exercise (postural reeducation, gentle gymnastics, range of motion exercises); (5) mind-body exercises (Tai Chi, Qigong, Baduanjin, Yoga, Alexander techniques, Feldenkrais); (6) exergames / video games / virtual reality (active game play, Wii games, Kinect games); and (7) multi-component exercises (combine two or more categories described above with or without a cognitive component).

We will extract information on intervention characteristics related to psychological / social / cognitive aspects:

- *Cognitive demand of exercise* categorized as either low or high and assesses by their affiliation to a specific type of exercise. Low cognitive demanding exercises include aerobic exercises and strength exercises / resistance exercises. High cognitively demanding exercises include balance exercises, coordination exercises, mind-body exercises, exergames, and cognitive tasks.

- *Social interaction* categorized by according to whether exercise sessions are conduced (1) in a group-based fashion, (2) an individual-based session containing only verbal interactions, and (3) an individual-based fashion containing exercising with partner(s) in cooperation and / or opposition [38].

- *Supervision during exercise practice* categorized as 1) home based, 2) supervised by a professional, and 3) not specified. Within home-based programs, participants engage in the program independently, relying on written or video materials for guidance but lacking real-time feedback during exercise. In supervised programs, a professional coach or instructor provides guidance and feedback to participants throughout the sessions.

- *Behavioral change techniques* categorized as follows: (1) no behavioral change technique; (2) goals and planning; (3) feedback and monitoring; (4) social support; (5) shaping knowledge, (6) natural consequences; (7) comparison of behavior; (8) associations; (9) repetition and substitution; (10) comparison of outcome; and (11) rewards and threat [42].

- *Compliance* assessed as percentage of sessions attended.

We will further extract information on study characteristics related to study design and statistical analysis aspects:

- *Study design* of the RCT categorized as either (1) parallel-group design or (2) cluster design.

- *Type of analysis* used by the authors to test the effect of the intervention categorized as follows: (1) Per-Protocol analysis (PP) where participants were excluded from final analysis due to low adherence rate and that may introduce a type 1 error; (2) Intention-To-Treat analysis (ITT) where missing data are replaced with different imputation methods and that increase the likelihood of type II errors; (3) analyses where all available data were considered, such as Complete-Case analysis (CC) where participants with missing data were excluded from final analysis, ITT where different sample sizes at baseline and at the end of the intervention were considered for final analysis, or analysis of RCTs with no missing data

- *Dropout rate* as the number of participants who were either excluded from the final analysis due to low adherence rates (PP) or dropped out during the study (CC and ITT) for each exercise and control group arm.

- Type of control group categorized as either (1) passive or (2) active. Passive control groups can be further subdivided to (1a) usual care / treatment as usual, (1b) habitual activities, (1c) waiting list. Active control groups can be further subdivided into (2a) health education, (2b) non-physical leisure activities, (2c) stretching / very light exercise / relaxation, (2d) social interaction.

## Individual characteristics

Concerning individual characteristics, we will extract information on the following moderator variables:

- *Age* assessed in years.

- *Sex* assessed as percentage of women.

- *Level of education* at baseline assessed as years of education.

- *Functional capacity* at baseline assessed with the Hand Grip Strength Test and the 6-min walk test. We acknowledge the existence of additional suitable indicators for functional capacity, such as $VO_2$max. However, in the process of selecting the individual characteristic variables, we followed a similar approach as for the secondary outcomes, selecting only those where enough studies (more than three out of a total of approximately 100 studies) reported a valid measure for a given characteristic.

- *Global cognition* at baseline assessed with the Mini Mental State Examination (MMSE) [99] or the Montreal Cognitive Assessment (MoCa) [100].

- *Population* categorized as follows: (1) neurodegenerative diseases (e.g., Alzheimer disease, Parkinson disease, dementia); (2) cardiovascular diseases (e.g., heart failures, strokes, hypertension); (3) other diseases (e.g., osteoarthritis, osteoporosis, diabetes, cancer, hemodialysis,); (4) DSM diagnosis of depressive disorders (e.g., minor or major depression); (5) cognitive impairments (e.g., mild cognitive impairment, cognitive frailty, major neurocognitive disorder); (6) other disabilities (visual impairment, frailty, wheelchair); and (7) no specific disease / disorder / disability (e.g., healthy older adults, sleep complaints, pre-frailty, persons with high risk of falls).

## Data analysis

**Outcome measurement.**    For continuous data the standardized mean difference (SMD) as Cohen's d and Hedges' g together with the 95% confidence interval (CI) will be calculated. If dichotomous data is obtained, the risk ratio (RR) will be calculated with a 95% CI.

**Dealing with missing data.**    If data or statistics (e.g., standard deviations) are missing, we will contact the authors. If we are unable to obtain the missing information, the missing statistics will be handled according to the Cochrane guidelines [101] by analyzing only the available data (i.e., ignoring the missing data). Missing data due to study dropout will be considered as part of the risk of bias assessment, noted in the summary table and considered in the discussion of the results.

**Assessment of heterogeneity.**    We will visually inspect confidence intervals on forest plots for the degree of overlap to determine the potential direction and magnitude of heterogeneity.

We will quantify heterogeneity using Q (also known as $\chi^2$, or Chi$^2$) and $I^2$ ([[Q -df] / Q]*100% = variance$_{between}$ / variance$_{total}$ * 100%) [101]. Q describes whether observed differences in results are compatible with chance alone (without considering the power of the specific studies). $I^2$ describes the percentage of the variability in effect estimates that is due to heterogeneity rather than chance and is a measure for the impact of the observed heterogeneity on the meta-analysis.

**Assessment of reporting publication bias.**  To assess publication bias, we will visually inspect the funnel plot for asymmetry and apply statistical tests such as Egger's test [102]. If publication bias is detected, a sensitivity analysis will be performed to check whether the results are robust to this bias [70]. Other publication bias methods will be also used [70].

**Assessment of evidence and study quality.**  Quality / certainty of evidence will be assessed with the GRADE framework (Grading of Recommendations, Assessment, Development, and Evaluations [103]) and the Cochrane Collaboration risk-of-bias tool for RCTs (RoB2) [104, 105]. Pairs of two reviewers will be involved in the quality assessment. The two reviewers will independently rate the quality of the included studies with ROB2. The GRADE framework will be used to assess the evidence accumulated for each outcome overall and for the specific domains: 1) risk of bias, 2) imprecision, 3) inconsistency, 4) indirectness, 5) publication bias. Disagreements will be resolved by a third reviewer.

**Data synthesis.**  We aim to calculate summary effects for the outcomes. As it is expected that there will be considerable heterogeneity, a random-effects model will be used for the meta-analysis. Summary effects will be calculated as Hedges' g using random-effects weights. If data cannot be pooled between intervention studies (e.g., an insufficient number of RCTs investigating the respective outcome) findings will be reported in narrative syntheses for each outcome variable. All analyses will be performed using the software Comprehensive Meta-Analysis 4.0 [106] and R [107].

**Subgroup analysis and meta-regression.**  One of the present review's hypothesis is that chronic exercise interventions differ in their effectiveness in reducing overall depression severity and the different symptoms of depression in older adults across participants with different individual's characteristics (i.e., age, sex, education, functional capacity, global cognition, population) and across interventions with different exercise characteristics (i.e., frequency, intensity, duration, type of exercise, cognitive demand, social interactions, exercise supervision, behavioral change techniques, compliance, study design, dropout-rate, type of control group). As such, to determine how different individual's and exercise characteristics / variables play a role, we will conduct subgroup analysis for categorical variables and meta-regression for continuous variables.

**Sensitivity analysis.**  We plan to carry out sensitivity analysis for the primary outcomes to assess the extent to which the results are robust to methodological assumptions and decisions that were made when carrying out the synthesis, such as (1) the effect of study quality, (2) the effects of fixed vs. random effects models, (3) the effects of potential publication bias.

## Ethics and dissemination

The present study will use published data and does not require ethics approval.

## Discussion and conclusion

In this article, we have outlined a protocol for a systematic and meta-analytical review on the effects of chronic exercise on depression in older adults that is consistent with current best practice guidelines. Any deviations from this protocol will be indicated in the final manuscript. Dissemination will be through presentations at national and international conferences and through a robust peer-reviewed publication.

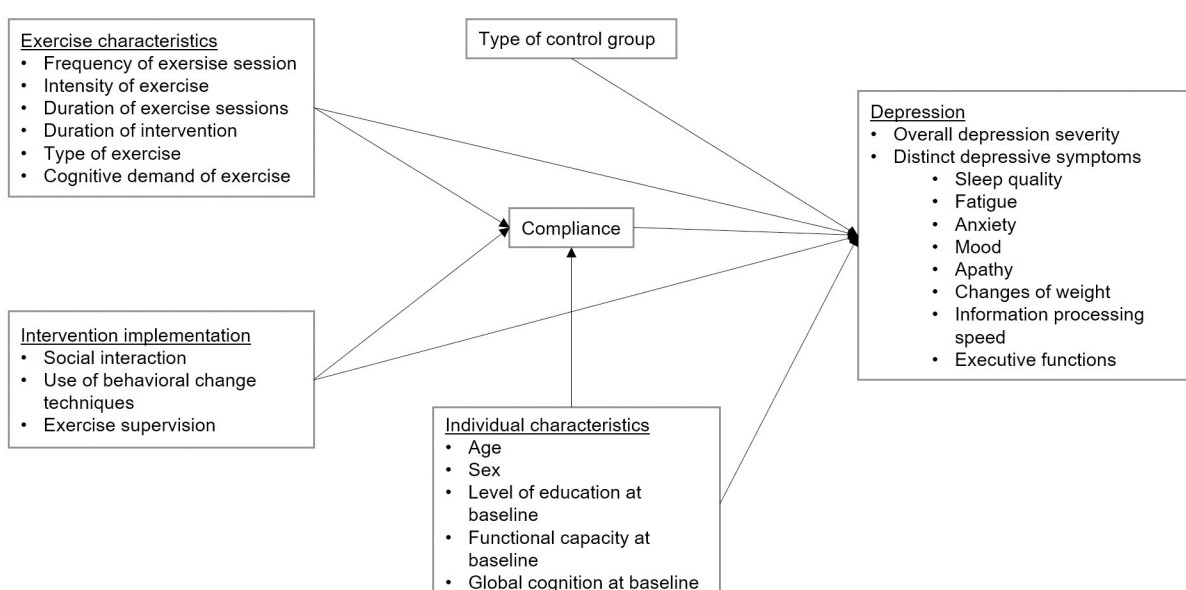

**Fig 2. Working model of the role of intervention characteristics and individual characteristics in exercise interventions effects on late-life depression.**

Our meta-analytical review will be the first that examines the effects of chronic exercise on different symptoms of depression in older adults and as a function of a large variety of moderators depending on intervention characteristics and individual characteristics. In line with the already published meta-analyses, we expect to find a moderate effect of exercise on depression compared to active and passive control groups with no exercise components [49]. We further expect the results to be moderated by intervention characteristics and individual characteristics (Fig 2).

The present review aims to make significant contributions to both research and clinical practice. Firstly, the conclusions drawn from this review may serve as a basis for developing evidence-based recommendations for individualized exercise programs that can help alleviate depression in older adults. For instance, it can help determine the exercise regimen in terms of the FITT principles. Secondly, the review results may reveal research gaps that can stimulate the formulation of research questions for further RCTs. For instance, it can reveal whether certain moderator variables, such as comorbidities or physical fitness level still require further research to understand their influence. Finally, this review aims to enhance the understanding of the effects of chronic exercise on various health aspects in older adults within the "Network on evidence-based physical activity in old age" (https://physagenet.eu/). In this network, several reviews on the impact of chronic exercise on different health aspects in older age will be carried out and the findings will be compared and consolidated. By aligning research in this network on similar protocols and assessment tools, gaps in terms of generalizability in the large body of research on physical exercise in old age will be filled.

Besides the above-mentioned strengths, the study has some limitations as well. Treatment of depression requires a multidimensional approach including pharmacological treatment and psychotherapy. We plan to exclude RCTs with tested effects of pharmacological treatment and / or psychotherapy. Therefore, no conclusions can be drawn for exercise interventions as add-on treatment. We are aware that by that we might miss some clinically relevant insights. We chose this approach to extract the "pure" impact of chronic exercise on depression in old age and to obtain results comparable to reviews of other aspects of health in old age that will be

conducted in the "Network on evidence-based physical activity in old age" [108]. We further anticipate limitations in form of heterogenous data and a wide variance in the reporting of the exercise characteristics (e.g., detailedness) and the individual characteristics. We aim to deal with those limitations by being cautious in interpreting the results.

In summary we believe that our review can, on the one hand, contribute to clinical praxis and research on chronic exercise in late-life depression and, on the other hand, contribute to the understanding of the influence of chronic exercise on various aspects of health in old age.

## Supporting information

**S1 Checklist. PRISMA-P 2015 checklist.**
(DOCX)

**S1 File. Search strategy.**
(PDF)

## Acknowledgments

We would like to acknowledge the support and resources provided by the all members of the EU COST Action CA20104—Network on evidence-based physical activity in old age (PhysA-geNet), whose collaboration is crucial for this study. The complete membership list of the consortium is available here: https://www.cost.eu/actions/CA20104/#tabs+Name:Working%20Groups%20and%20Membership.

## Author Contributions

**Conceptualization:** Melanie Mack, Christoforos D. Giannaki, Antonia Kaltsatou, Claudia Voelcker-Rehage, Michel Audiffren.

**Data curation:** Melanie Mack, Andreea Badache, Arzu Erden, Christoforos D. Giannaki, Sandra Haider, Antonia Kaltsatou, Burcu Kömürcü Akik, Yaël Netz, Iuliia Pavlova, Pinelopi S. Stavrinou, Claudia Voelcker-Rehage, Michel Audiffren.

**Funding acquisition:** Yaël Netz.

**Investigation:** Melanie Mack, Andreea Badache, Arzu Erden, Christoforos D. Giannaki, Sandra Haider, Antonia Kaltsatou, Burcu Kömürcü Akik, Yaël Netz, Iuliia Pavlova, Pinelopi S. Stavrinou, Claudia Voelcker-Rehage, Michel Audiffren.

**Methodology:** Melanie Mack, Antonia Kaltsatou, Michel Audiffren.

**Project administration:** Melanie Mack, Christoforos D. Giannaki, Claudia Voelcker-Rehage, Michel Audiffren.

**Software:** Melanie Mack, Claudia Voelcker-Rehage, Michel Audiffren.

**Supervision:** Melanie Mack, Christoforos D. Giannaki, Claudia Voelcker-Rehage, Michel Audiffren.

**Validation:** Melanie Mack, Christoforos D. Giannaki, Claudia Voelcker-Rehage, Michel Audiffren.

**Visualization:** Melanie Mack, Michel Audiffren.

**Writing – original draft:** Melanie Mack.

**Writing – review & editing:** Andreea Badache, Arzu Erden, Christoforos D. Giannaki, Sandra Haider, Antonia Kaltsatou, Burcu Kömürcü Akik, Yaël Netz, Iuliia Pavlova, Pinelopi S. Stavrinou, Claudia Voelcker-Rehage, Michel Audiffren.

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
