## [Decision Letter · Decision Letter 0]

11 Sep 2023

PONE-D-23-17828The effects of chronic exercise on symptoms of depression in older adults: A protocol of a systematic and meta-analytic reviewPLOS ONE

Dear Dr. Mack,

Thank you for submitting your manuscript to PLOS ONE. After careful consideration, we feel that it has merit but does not fully meet PLOS ONE’s publication criteria as it currently stands. Therefore, we invite you to submit a revised version of the manuscript that addresses the points raised during the review process.

The manuscript presents a well-structured protocol that is both well-written and pertinent to the field of geriatric psychiatry and exercise science. I concur with the peer reviewer's assessment that the study could benefit from further refinements to enhance its overall quality.

**Changes favoring the Acceptance of the manuscript:**

Clarification of Rationale and Objective: Please make it clear how your review adds to the existing body of literature, especially with respect to the symptom-by-symptom analysis.

Methodological Details: Address the line-specific comments made by the reviewer, particularly those that require citation or clearer definitions.

Data Analysis: Clearly state how you plan to differentiate symptom-by-symptom, and how this will be supported methodologically.

Secondary Outcomes: Consider including symptoms like suicidal ideation and feelings of worthlessness as recommended by the reviewer.

Long-term Effects: While not mandatory, an exploration of the long-term effects of exercise on depression could enrich the study.

In general, please implement all the points raised by the peer reviewers, as requested.

We look forward to receiving your revised manuscript.

Kind regards,

Alessandro Rodolico

Academic Editor

PLOS ONE

Journal Requirements:

"This publication is based upon work from EU COST Action CA20104 - Network on evidence-based physical activity in old age (PhysAgeNet), supported by COST (European Cooperation in Science and Technology). https://www.cost.eu/, " ext-link-type="uri" xlink:type="simple">https://physagenet.eu/."

"This publication is based upon work from EU COST Action CA20104 - Network on evidence-based physical activity in old age (PhysAgeNet), supported by COST (European Cooperation in Science and Technology)”."

"This publication is based upon work from EU COST Action CA20104 - Network on evidence-based physical activity in old age (PhysAgeNet), supported by COST (European Cooperation in Science and Technology). https://www.cost.eu/, https://physagenet.eu/. The funders did not and will not have a role in study design, data collection and analysis, decision to publish, or preparation of the manuscript."

"All the authors of this manuscript are members of the COST network PhysAgeNet (CA 20104). The authors have declared that no competing interest exists."

Additional Editor Comments:

The protocol is well-written and relevant, but it could benefit from refinements in key areas such as rationale and objective, methodological details, secondary outcomes, individual characteristics, and data analysis. I recommend addressing the peer reviewer's comments to improve the manuscript overall.

Reviewers' comments:

Reviewer's Responses to Questions

**Comments to the Author**

1. Does the manuscript provide a valid rationale for the proposed study, with clearly identified and justified research questions?

Reviewer #1: Partly

Reviewer #2: Yes

2. Is the protocol technically sound and planned in a manner that will lead to a meaningful outcome and allow testing the stated hypotheses?

Reviewer #1: Yes

Reviewer #2: Yes

3. Is the methodology feasible and described in sufficient detail to allow the work to be replicable?

Reviewer #1: Yes

Reviewer #2: Yes

4. Have the authors described where all data underlying the findings will be made available when the study is complete?

Reviewer #1: Yes

Reviewer #2: Yes

5. Is the manuscript presented in an intelligible fashion and written in standard English?

Reviewer #1: Yes

Reviewer #2: Yes

6. Review Comments to the Author

You may also provide optional suggestions and comments to authors that they might find helpful in planning their study.

Reviewer #1: This systematic review protocol is generally well-written. However, there are some topics that I think should be addressed. Also, I failed to understand how this review is different from the already existing ones on the topic. Is it by analyzing symptoms of depression depression (MDD and/or dysthimia I guess) one by one? It is not clear if this analysis is possible and how it will be performed.

More specific comments are below:

l. 65-67. Please refer to the year of this estimates.

l. 69. Is aging associated with lower socioeconomic status?

l. 79-84. These sentences need to be supported by references.

l. 82. What is medical morbidity?

l. 91-93. What you mention as adverse events are considered side (secondary) effects. Normally adverse events in depression are suicide death, suicide attempt and relapse.

l. 98. “which can be easily applied independently” what do you mean by this?

l. 135-136. I did not understand this sentence.

l. 136-138. This sentence needs to be supported by references.

l. 192-194. This sentence needs to be supported by references.

l. 209-212. What would be the benefit of doing this? Also, how are you going to discriminate symptom by symptom?

l. 251-255. How is your review not going to fall in this same issue?

l. 298-300. By considering mind-body exercises (yoga, tai chi) and exercises with cognitive therapy how are you going to be sure that the effect is due to exercise itself and not the other components of this kind of interventions?

l. 330. Why are you not using PubMed or Medline or EMBASE? Is it included in any other?

l. 371-374. What is the rational for this? Why don’t you extract both mean changes?

l. 380. I am not understanding the secondary outcomes section. Are you defining the secondary outcomes that will be extracted (if they appear in the RCT) and the measures accepted by you? E.g., for sleep quality, if it is not assessed by either PSQI or ISI you will not extract that information?

l. 432. What is very light PA?

l. 519. What are the parameters GRADE will be assessing? Only the overall effect?

l. 534. Where are these hypotheses stated?

Reviewer #2: I would like to commend the authors on their work in presenting the protocol of a review article that focuses on exploring the impact of chronic exercise on depressive symptoms among older adults. The study's objective to delve into the effects of exercise on specific symptoms of depression, in addition to its overall severity, is a noteworthy contribution to the existing literature. The incorporation of various moderators and techniques to correct for publication bias adds sophistication to the protocol. While the manuscript is well-written and the protocol demonstrates a commendable level of detail, I do have a few suggestions that I believe could further enhance the overall quality of the study.

Rationale and Objective:

I appreciate the authors' intention to investigate different intervention characteristics as potential moderators. One aspect to consider is the distinction between supervised and unsupervised exercise treatments, which could be an important moderator. Incorporating supervision as a potential moderator could provide valuable insights. If the authors do not include supervision as moderator, please elaborate.

Method: Intervention:

The manuscript's title indicates a focus on chronic exercise, but the defined intervention duration of at least one week (with more than one exercise session) can be perceived as relatively short for a chronic exercise context. The focus on chronic exercise in the title is perhaps misleading.

Method: Secondary Outcomes:

The authors' intent to assess the effects of chronic exercise on specific depression symptoms is commendable. However, it's worth considering the inclusion of symptoms like suicidal ideation and feelings of worthlessness (low self-esteem; e.g. Kandola et al., 2019). Incorporating these symptoms in the review could provide a more comprehensive overview of the subject matter.

Additionally, I'd like to highlight the authors' approach to evaluating the symptom "significant weight loss when not dieting or weight gain" using BMI. Given that regular exercise might influence BMI changes, it's crucial to address the potential for misinterpretation, where weight changes could mistakenly signify loss of appetite or involuntary weight loss. The authors are encouraged to discuss how they plan to handle this potential issue.

Method: Individual Characteristics:

The introduction's mention of physical fitness as a potential moderator of the exercise-depression relationship is noteworthy. I suggest considering the inclusion of baseline VO2max as an indicator of physical fitness, if such data is available, alongside the hand grip strength test and the 6-min walk test. Otherwise please indicate why you did not include VO2max as moderator.

Data Analysis:

The authors' choice to report the standardized mean difference from pre to post-treatment is appropriate. To further enrich the study, it might be valuable to explore and report any potential long-term effects of exercise, including whether participants maintained their exercise routines over time.

7. PLOS authors have the option to publish the peer review history of their article (what does this mean?). If published, this will include your full peer review and any attached files.

Reviewer #1: No

Reviewer #2: No

---

## [Author Response · Author response to Decision Letter 0]

12 Oct 2023

The reviewers’ comments were highly insightful and enabled us to improve the quality of our manuscript. Our revisions to the text are recorded using yellow highlighted font in MS Word. You can find our point-by-point responses to the reviewers’ comments in the document “response to reviewers”.

---

## [Decision Letter · Decision Letter 1]

3 Jan 2024

Chronic exercise effects on overall depression severity and distinct depressive symptoms in older adults: A protocol of a systematic and meta-analytic review

PONE-D-23-17828R1

Dear Dr. Mack,

We’re pleased to inform you that your manuscript has been judged scientifically suitable for publication and will be formally accepted for publication once it meets all outstanding technical requirements.

Kind regards,

Claudio Damião Rosa, M.D

Academic Editor

PLOS ONE

Additional Editor Comments (optional):

Reviewers' comments:

Reviewer's Responses to Questions

**Comments to the Author**

1. Does the manuscript provide a valid rationale for the proposed study, with clearly identified and justified research questions?

Reviewer #1: Yes

Reviewer #2: Yes

2. Is the protocol technically sound and planned in a manner that will lead to a meaningful outcome and allow testing the stated hypotheses?

Reviewer #1: Yes

Reviewer #2: Yes

3. Is the methodology feasible and described in sufficient detail to allow the work to be replicable?

Reviewer #1: Yes

Reviewer #2: Yes

4. Have the authors described where all data underlying the findings will be made available when the study is complete?

Reviewer #1: Yes

Reviewer #2: Yes

5. Is the manuscript presented in an intelligible fashion and written in standard English?

Reviewer #1: Yes

Reviewer #2: Yes

6. Review Comments to the Author

You may also provide optional suggestions and comments to authors that they might find helpful in planning their study.

Reviewer #1: Thank you for addressing all the comments adequately. Hope it helped.

Good luck with the systematic review.

Reviewer #2: The authors present the revised protocol of a review article that focuses on exploring the impact of chronic exercise on depressive symptoms among older adults. The authors adequately addressed all comments in a satisfactory manner.

7. PLOS authors have the option to publish the peer review history of their article (what does this mean?). If published, this will include your full peer review and any attached files.

Reviewer #1: No

Reviewer #2: No

---

## [Editor Report · Acceptance letter]

2 May 2024

PONE-D-23-17828R1 

PLOS ONE

Dear Dr. Mack, 

I'm pleased to inform you that your manuscript has been deemed suitable for publication in PLOS ONE. Congratulations! Your manuscript is now being handed over to our production team.

Kind regards, 

on behalf of

Professor Claudio Damião Rosa 

Academic Editor

PLOS ONE